# Microstructure and Properties of the Copper Alloyed with Ag and Ti Powders Using Fiber Laser

**DOI:** 10.3390/ma13112430

**Published:** 2020-05-26

**Authors:** Mariusz Krupiński, Paulina Ewelina Smolarczyk, Mirosław Bonek

**Affiliations:** Department of Engineering Materials and Biomaterials, Silesian University of Technology, Konarskiego 18a, 44-100 Gliwice, Poland; paulina.smolarczyk@polsl.pl (P.E.S.); miroslaw.bonek@polsl.pl (M.B.)

**Keywords:** non-ferrous metal, copper, alloying, laser surface modification, microstructure

## Abstract

The scope of the work covers the development of the relationship between the chemical composition of surface-modified copper and the diffusion of alloy elements as well as the microstructure and mechanical properties. This article presents the impact of laser alloying with titanium and silver powders on the microstructure and mechanical properties of copper. In order to investigate the phenomena occurring during the laser alloying process, microstructural studies were performed using scanning electron microscopy (SEM), optical microscopy, and energy dispersive x-ray spectroscopic (EDS) analysis of the chemical composition in micro-areas. In addition, to test the properties of the resulting alloy, abrasion resistance, hardness measurement at low loading force, and conductivity measurements were performed. As a result of alloying with Ag and Ti powders, three distinct zones were indeed recognized: re-melting zone (RZ), diffusion zone (DZ), and heat affected zone (HAZ). The surface modification that results from laser alloying increases the hardness as well as the abrasion resistance of the material. Overall, it was found that laser alloying with Ti powder increased the strength of the copper surface layer due to the formation of intermetallic phases (Cu_3_Ti_2_). It was also found that laser alloying with Ag powder changed the mechanical properties of the surface layer due to the solid solution strengthening.

## 1. Introduction

It is well-known that copper has good conductivity properties, although it lacks significant strength. Strengthening of copper is most often obtained by plastic deformation, however, this method has a disadvantage in the fact that at elevated temperatures, recrystallization occurs, which decreases strengthening. Other methods require chemical composition modification and concern the strengthening of copper alloys and include dispersion, precipitation, solid solution strengthening, or strengthening through intermetallic phases [1,2,3].

There are various other methods to improve the mechanical properties of copper such as alloying. Research aimed at improving the material structural quality, while maintaining an electrical conductivity above 57 MS/m has been carried out. For example, CuB2 and CuZr30 grain refinements have been used to modify certain materials. The test results showed that the modification of the chemical composition caused grain refinement and also maintained the electrical conductivity [4]. Copper alloys such as Cu–Ni–Si and Cu–Cr, modified with rhenium and silver to improve mechanical properties, have also been investigated and it was discovered that supersaturated, plastically deformed, and aged alloys showed increased mechanical properties in these types of alloys [5,6].

It is also possible to use plastic deformation to change the mechanical properties of copper. Specifically, increased abrasion resistance is obtained following large-scale copper plastic deformation. In general, coarse-grained copper has been shown to have better abrasion resistance [7], however, there was no significant increase in electroconductivity at the obtained grain refinement compared to the coarse-grained structure.

Grain refinement due to plastic deformation may be reduced in recrystallization, and copper is often employed at elevated temperatures that promote such recrystallization processes. During copper modification, the side effect of increasing mechanical properties is a decrease in conductivity [8]. It is important that in the main copper supply, the current flows entirely on the surface of the cable (i.e., the so-called skin effect in electrical engineering). Considering that a copper surface is prone to oxidation, this resistance may be much higher than that resulting from the conductivity value of the pure bulk metal [8,9].

Modification of the chemical composition with alloying elements can be particularly helpful when altering the surface layer. It allows for adapting mechanisms that occur in the alloy for use in surface modifications, although, as a rule, cooling speeds in the latter processes are much faster. In [10,11], it was found that the occurrence of Ag phases as well as CuTi_2_, Cu_3_Ti_2_, and Cu_4_Ti_3_ are possible. As soon as the eutectic Ag–Cu alloy melts (melting point (M.P.) = 780 °C), the Ti begins to dissolve in the liquid. When the Ti activity in the liquid is sufficiently high near the Ti surface, the Cu_4_Ti compound nucleates and rapidly forms a continuous reaction layer of small crystals that extracts Ti from the liquid. Other Ti–Cu compounds can then nucleate and grow via solid-state reaction-diffusion between Ti and the Cu_4_Ti layer. A reaction zone is thus formed (Figure 1a), consisting of four dense, single-phased layers of compounds arranged according to the diffusion path (Figure 1b) in the Cu–Ag–Ti phase diagram (see Figure 2) [12,13].

The modification of the surface layer of metals and alloys with a laser beam is used in a number of surface treatments such as alloying, melting, and re-melting. In the cases of alloying, feathering, and melting, it is also possible to change the chemical composition of the created layer. Various other effects including homogenization, grain refinement, and increased hardness of the surface layer are usually obtained as a result of re-melting. Re-melting treatments generally induce a tensile stress that is associated with the heating and cooling processes of the improved layer [14,15]. The stress value depends on the parameters of the laser treatment (i.e., the radiation power density); the method of laser processing, whether it is a single transition or multi-path treatment; the degree of covering the paths; the type of material; any phase changes occurring during the heat treatment; and finally, the adopted measuring method [16,17,18].

During laser surface modification of metals and their alloys, various types of lasers can be used. For example, Nd: YAG (neodymium-doped yttrium aluminum garnet) lasers (λ = 1064 nm, 2.5 kW) are employed to increase corrosion and abrasion resistance [19], and they can be applied at lower laser power (λ = 1064 nm, 2 kW) to reduce hardening costs in low-alloy steels [20]. A CO_2_ laser (10 kW) is used for grain refinement, or to increase corrosion resistance or change the microstructure [21]. It was reported that a CO_2_ laser could be used for alloying with TiO_2_ and SiC powders to modify the microstructure and microhardness of aluminum [22]. Fiber lasers (λ = 1064 nm, 2 kW) are used to increase wear resistance and corrosion resistance [23], and interestingly, lasers of this type with increased power (8 kW) are used in the process of laser metal deposition [24]. Fiber lasers with a power of 4 kW and a wavelength of λ = 1070 nm have been used to increase the wear resistance and hardness of aluminum alloys as a result of microstructural changes [25].

Laser modification has also been shown to improve the strength of a Cu surface layer. In these experiments, a Ni-based powder was applied to a copper surface using an acetone solution of cellulose acetate as a binder. It was found that after modification, the sample hardness increased by a factor of six, and the electrical conductivity decreased slightly. The increase in mechanical properties has been attributed to the precipitation hardening of boride and carbide [26]. Laser surface treatment also has applications toward increasing the abrasive wear resistance of the treated material. Depending on the material used, a fine, more homogeneous structure can be obtained, which consequently leads to increased wear resistance [27].

Modification of copper for industrial applications is largely focused on increasing wear resistance, which contributes to increasing the service life of copper electrical contacts [28]. In laser beam processes, the selection of system parameters is very important (e.g., if the laser beam power density is too low or the scanning speed is too high, the structure of the resulting melted layer may be heterogeneous) [29].

Laser surface modification is a common technique that is used mainly with oxides and carbides. The use of ceramic powders has many advantages including structural stability, even at elevated temperatures. The disadvantages of this method include (i) wettability, (ii) conglomerate formation, (iii) the large amount of powder that must be introduced into the melted zone, (iv) the powder’s morphology and size, and (v) the melting and subcooling of powders (mainly carbon component) into solution. The use of pure metals may cause strengthening as a result of the formation of intermetallic phases, precipitation, and solid solution mechanisms. A material with high mechanical properties including creep resistance, and high conductivity of the surface layer can be applied in contact–friction joints, commutators, sliding contacts, or electrode tips, which was the reason for using Ag and Ti to modify the laser surface of copper.

## 2. Materials and Methods

In order to investigate the effect of the laser modification of the copper surface layer with Ti and Ag powders on its microstructure and properties, the following steps were undertaken:99.95% pure, drawn flat copper bar (M1Ez4) delivered by the Kafra Color Metals Company (Warsaw, Poland) was used. Dimensions: length = 80 mm; width = 30 mm; thickness = 10 mm. Samples were ground with sandpaper #1200.Titanium powder (purity 99.5%, −325 mesh) and silver powder (purity 99.9%, −325 mesh) delivered by Alfa Aesar (Kandel, Germany) were applied to the surface of the copper.Alloying of the surface layer was performed with the fiber laser, Ytterbium Laser System, YLS-4000-S2T (IPG Photonics Corporation, Oxford, MA, USA; λ = 1070 nm, maximum laser beam power = 4000 W). Various process parameters were tested, but the best results were achieved for the parameters given in Table 1. Helium was the protective gas during the process. Twenty samples with were performed with a laser. The re-melting width was 2–3 mm and length was 38–42 mm (Figure 3).Zeiss Supra 35 scanning microscope (SEM, Thornwood, New York, NY, USA) was used to study the microstructure using the secondary electron method. The EDS technique (energy-dispersive x-ray spectroscopy) was used for the quantitative and qualitative analysis of micro-areas. Directly before the process, the treated surface was mechanically ground (15 mm) and cleaned with methyl alcohol and compressed air.Metallographic analysis was performed using optical microscopy on an Axio Observer from Zeiss (Thornwood, New York, NY, USA). After grinding, the laser-alloyed copper was subjected to polishing and electrolytic etching in D2 Struers Electrolyte (Thornwood, New York, NY, USA) (polishing: time = 10 s, voltage = 10 V; etching: time = 3 s, voltage = 2 V; temperature = 24 °C) on a Struers LectroPol-5 device (Thornwood, New York, NY, USA). Observations were made in bright field and in polarized light.Tests were carried out using a copper cathode (wavelength = 1.5406 Å) on a Panalytical MPD X’Pert Pro device (Egham, UK), enabling the determination of the interplanar distance and phase composition after modification.Vickers hardness method [HV1] was measured on a Future Tech FM-ARS 900 device (FM-ARS9000, Future-Tech, Tokyo, Japan); process parameters: t = 12 s, F = 9807 N. Five hardness measurements were taken for each sample. Measurements were made on the copper surface along each alloying.Conductivity tests were performed on a Sigmatest Foerster 2.069 (FOERSTER, Pittsburgh, PA, USA) device, which measures the electrical conductivity based on the complex impedance of the probe. Measurements were carried out at ambient temperature with a device operating frequency of 60 kHz. Before measuring the samples, the device was calibrated on a set of two standards with electrical conductivity values of 4.4 MS/m (8% IACS) and 58 MS/m (100% IACS). Samples were ground with sandpaper with a grain size of 3 µm.Abrasion resistance was tested on the Tribometer CSM Instruments device by the ball-on-plate method using a ceramic ball (Al_2_O_3_) with a 6 mm diameter as a counter-sample; dry test environment at ambient temperature. Parameters: full amplitude = 6 mm, linear speed = 2 cm/s, counter-sample load = 5 N, sliding distance = 25 m, frequency = 1.06 Hz. Reciprocating linear motion. Samples were ground with sandpaper with grain size #2400.Using the Taylor-Hobson Surtronic 25 contact profilometer (Taylor Hobson Ltd., Leicester, UK), the wear track profile was tested with the following parameters: sampling line = 1.25 mm, wipe path = 6 mm. Resolution at 10 µm range–0.01 µm, resolution at 100 µm range–0.1 µm. Based on the wear profile field and the length of wear tracks, the volume of worn material was determined.

## 3. Results and Discussion

Modification of a copper surface with silver powders using laser techniques resulted in the segregation of three zones, namely, the re-melting zone (RZ), the diffusion zone (DZ), and the heat affected zone (HAZ), as shown in Figure 4. As in the Ag case, titanium powder alloying also resulted in the creation of similar zones (Figure 5). The re-melting zones were measured to be 392.67 µm for copper alloyed with silver powder, and 439.7 µm for titanium alloyed copper.

Depending on the powders used and the heat dissipation rate, different processes occur in the visible re-melting zone. In the solidified zone, part of the silver was dissolved in the α matrix, while, due to segregation, its excess was released in the form of Cu + Ag globular eutectics, which can be seen in Figure 6. The EDS analysis of the micro-areas marked in Figure 6a, is shown in Figure 6b–d (Table 2). Some of the silver (about 9 wt.%) was dissolved in the α matrix (Figure 6a) as a result of the solidification of the re-melted zone (RZ) (Figure 4), and because of the mass of the samples, the cooling rate was large enough that the excess silver remained dissolved in the matrix. When the excess silver was pushed out by the crystallization front, this led to the micro-segregation and coagulation of Cu + Ag eutectic at the α phase boundary (interdendritic spaces). The size of the remaining eutectics was between 2–3 µm in diameter. Therefore, due to this alloying, the mechanical properties of the material including hardness were increased.

The re-melting zone of the copper alloyed with titanium powder showed directional solidification (toward the front of heat dissipation) as shown in Figure 7 and Figure 8. In the case of solidification of the re-melted zone (RZ), directional crystallization caused the copper samples modified with titanium powder to adopt a structure consisting of an α matrix and a separated Cu–Ti phase. EDS testing of the re-melted zone area revealed that the mass composition was Ti 6.58% wt. and Cu 93.42% wt., which was confirmed by the intensity plot in Figure 9b. This was also verified by the scanning electron microscopy (SEM) study and distribution of the linear distribution of elements (Figure 10). There was a visible increase in mass concentration Ti from the RZ zone (point 0) to the HAZ zone (point 85 µm).

For the samples used after plastic deformation, recrystallization began with the DZ (diffusion zone) and also took place in the HAZ (heat affected zone) (Figure 11). Recrystallization went deeper into the base material to a depth of 2–3 mm, and occurred over a width slightly larger than the area of the RZ zone for the sample sizes used in this research and for the laser parameters given in Table 1 in the Materials and Methods Section.

The changes in the chemical composition and phase composition that occurred as a result of the laser modification of copper with silver and titanium powder were examined by x-ray structural analysis, and the results are presented in Figure 12. The excess of silver, which has been isolated in the α-phase dendrite in the form of eutectics, is visible on the graph. In addition, regarding the copper substrate material after silver alloying, the diffractograms showed the presence of oxides. The notable shift of the Cu peak was caused by the dissolution of Ag in Cu. Similarly, the change in the intensity of the Cu peak after alloying titanium powder was the result of the dissolution of Ti in Cu and the creation of the Cu_3_Ti_2_ phase. However, the Cu_3_Ti_2_ phase peak width did not exclude the formation of other Cu–Ti phases, as confirmed by previous work [11,12,30,31]. As a result of introducing silver and titanium into copper, there was a change in the network parameter, and the calculated values are given in Table 3.

After observing the changes in the copper material’s microstructure after alloying with the silver and titanium powders, hardness measurements were carried out, and the results are presented in Table 4. Compared to the copper substrate before treatment, silver powder alloying and titanium powder alloying increased the hardness by 17.2 and 356.82 HV, respectively.

A ball-on-plate test was performed for the base material before and after alloying with silver and titanium powder in order to analyze how the laser modification of copper affected the surface layer properties of the material [32]. In addition, profilometer tests allowed for the calculation of the surface and volume wear of the copper material before and after alloying with the silver and titanium powders (Figure 13, Figure 14 and Figure 15). The average abrasive surface was 683.9 µm^2^ for pure copper, 595.9 µm^2^ for copper alloyed with silver powder, and 360.3 µm^2^ for copper alloy with titanium powder. The average volume of wear track was also calculated for copper (4,103,400 µm^3^), copper alloyed with Ag powder (3,575,657 µm^3^), and copper alloyed with Ti powder (2,161,650 µm^3^). The presence of Cu_3_Ti_2_ intermetallic phases increased the abrasion resistance of the copper surface layer laser-alloyed with titanium.

The results of the conductivity test showed that the specific electrical conductivity of the material tested before the alloying process was higher than either case after treatment. The average conductivity for copper was 58.35 MS/m. In the case of copper alloyed with silver powder, the material average conductivity was 42.29 MS/m (standard deviation was 0.85), where a decrease by 27.5% was observed.

## 4. Conclusions

Herein, we report the various impacts of laser modification of a copper surface with silver and titanium powder.

Following such a treatment, a re-melting zone (RZ), a diffusion zone (DZ), and heat affected (HAZ) zone were identified in the materials after alloying with either powder tested. The diffusion zone was created between the re-melting zone and the heat affected zone, and there was no gap between the RZ and HAZ zones because the re-melting zone solidified in the direction of the surface. Applying specific laser treatment parameters to a defined volume of the laser-processed material, the depth of the re-melting zone determined for the silver powder alloyed Cu was 392.67 µm, and for the titanium powder alloyed Cu, it was 439.7 µm.

In Ag alloyed copper samples, some of the silver remained dissolved in the α matrix. As a result of segregation, excess silver crystallizes in the form of a globular eutectic Cu + Ag (about 2–3 µm in diameter). The silver dissolved in the α matrix caused solid solution strengthening. Titanium powder alloying caused crystallization of the Cu_3_Ti_2_ phase in the direction of heat dissipation within the α matrix.

Overall, it was concluded that alloying the copper surface with a fiber laser leads to improvements in several strength properties. The hardness increased by 15% for the Ag case, and by 417% for the Ti case. Additionally, the material’s abrasion resistance was increased, especially, as a result of the formation of intermetallic phases created after laser alloying with Ti powder. There was a decrease in the volume of wear of 53.6% in the case of the alloying of Ti powder. At the same time, conductivity decreased to 27.5%.

## Figures and Tables

**Figure 1 materials-13-02430-f001:**
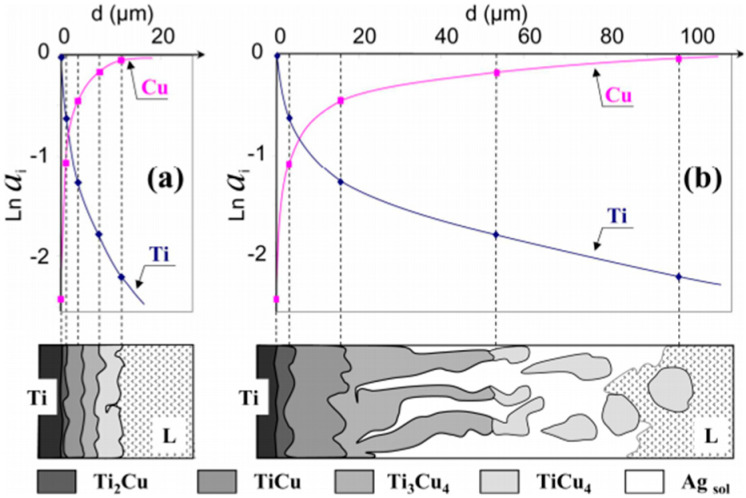
(**a**) Chemical activity of Cu and Ti in intermetallic phases formed as a result of the reaction at 790 °C in pure and (**b**) saturated 40% Ti–Ag in a liquid Cu alloy Reprinted with the permission of [12].

**Figure 2 materials-13-02430-f002:**
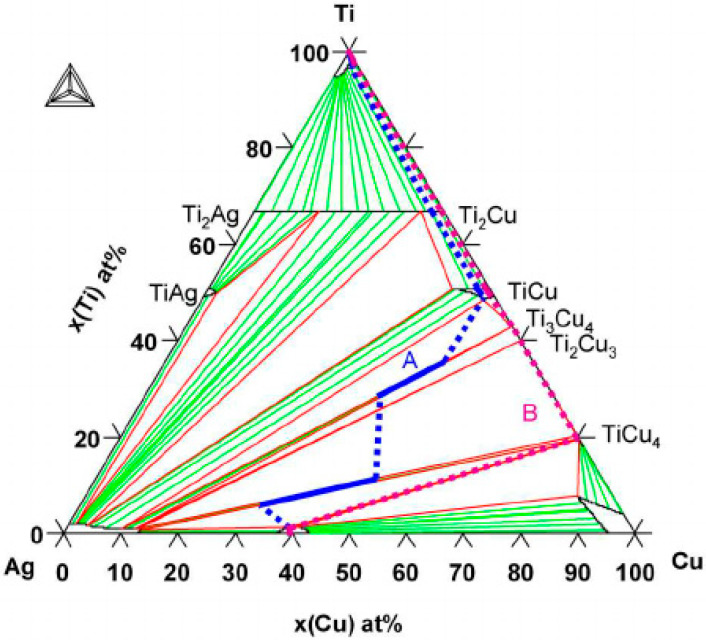
The Cu–Ag–Ti ternary phase diagram calculated at 790 °C. Two diffusion paths have been drawn corresponding to the reaction zones Reprinted with the permission of [12].

**Figure 3 materials-13-02430-f003:**
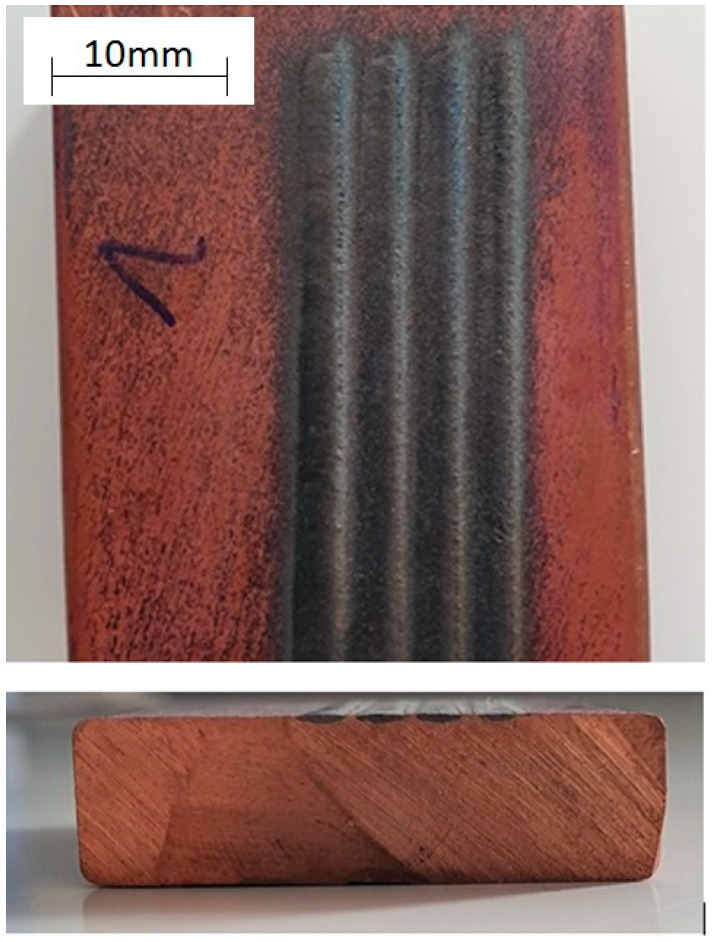
Sample of the melting’s surface and a cross-section before grinding with sandpaper.

**Figure 4 materials-13-02430-f004:**
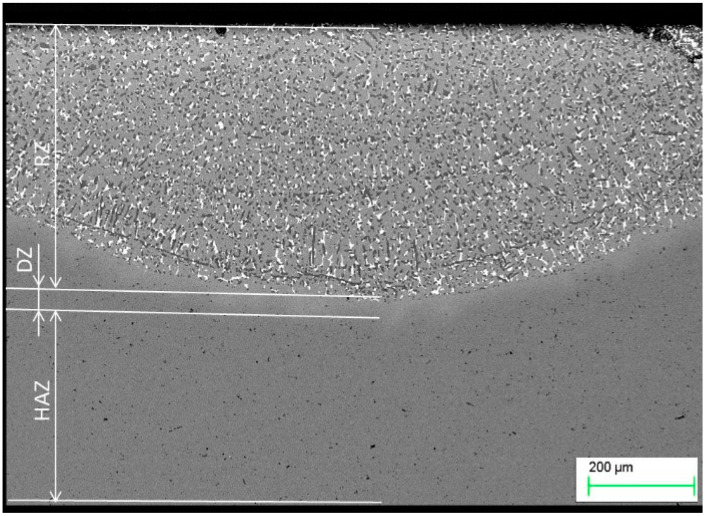
Microstructure of copper after Ag alloying. RZ = re-melted zone, DZ = diffusion zone, HAZ = heat affected zone.

**Figure 5 materials-13-02430-f005:**
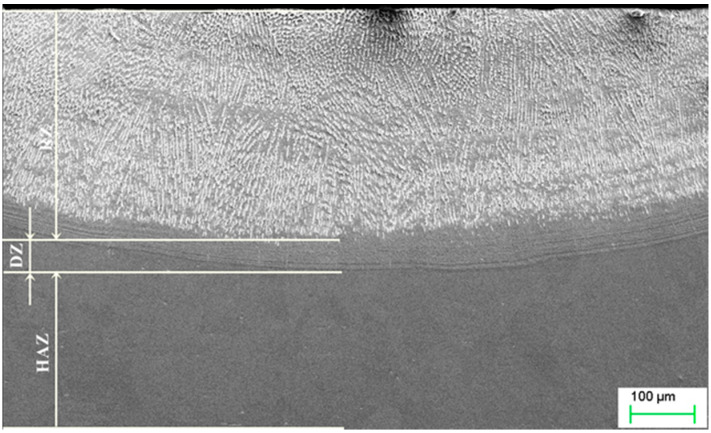
Microstructure of copper after Ti alloying. RZ = re melted zone, DZ = diffusion zone, HAZ = heat affected zone.

**Figure 6 materials-13-02430-f006:**
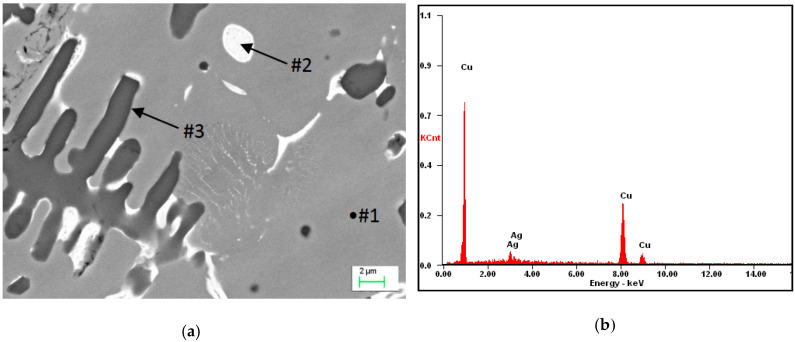
(**a**) Microstructure of Cu after alloying by Ag: #2 Cu + Ag eutectic; (**b**) analysis of energy dispersive x-ray spectroscopy (EDS) from the micro-area #1; (**c**) analysis of EDS from the micro-area #2; and (**d**) analysis of EDS from the micro-area #3.

**Figure 7 materials-13-02430-f007:**
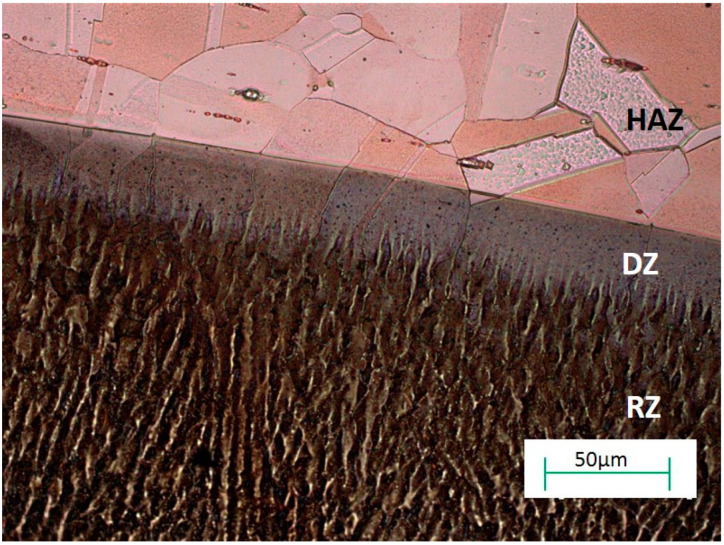
Optical microscopy image showing the microstructure of copper after Ti alloying.

**Figure 8 materials-13-02430-f008:**
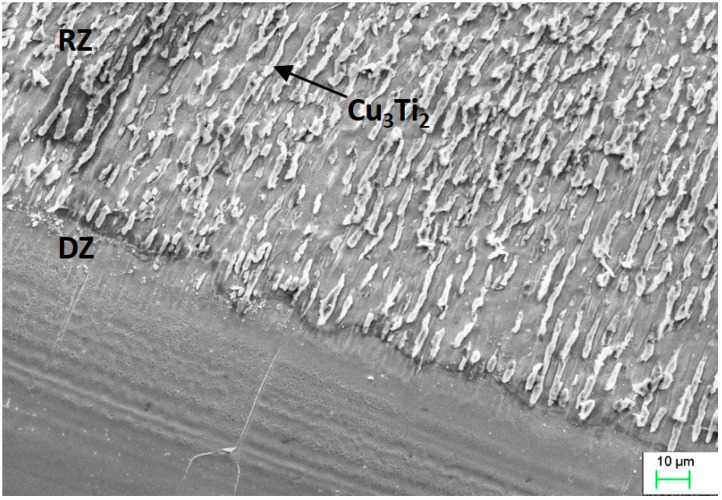
Scanning electron microscopy (SEM) image showing the microstructure of copper after Ti alloying.

**Figure 9 materials-13-02430-f009:**
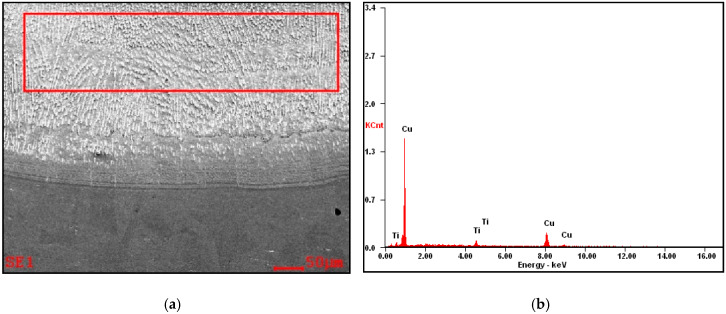
(**a**) The microstructure of the copper surface-alloyed with titanium, and (**b**) analysis of EDS from the selected area: Ti 6.58% wt., Cu 93.42% wt.

**Figure 10 materials-13-02430-f010:**
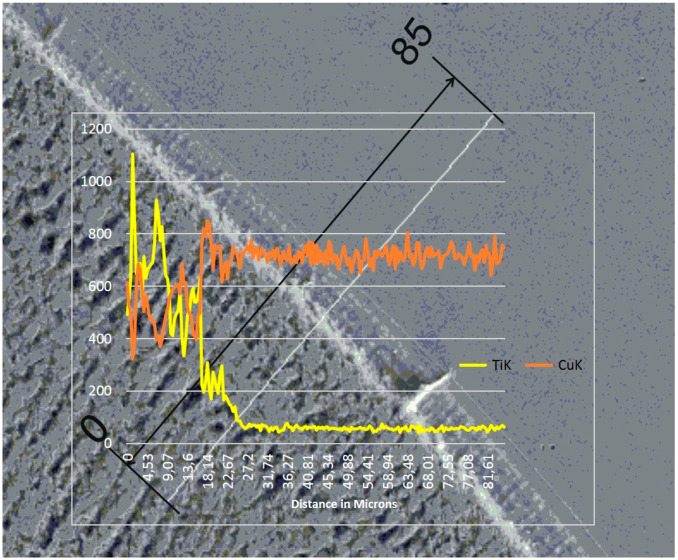
Linear chemical composition analysis at the zone boundary. The scale bar is 85 µm.

**Figure 11 materials-13-02430-f011:**
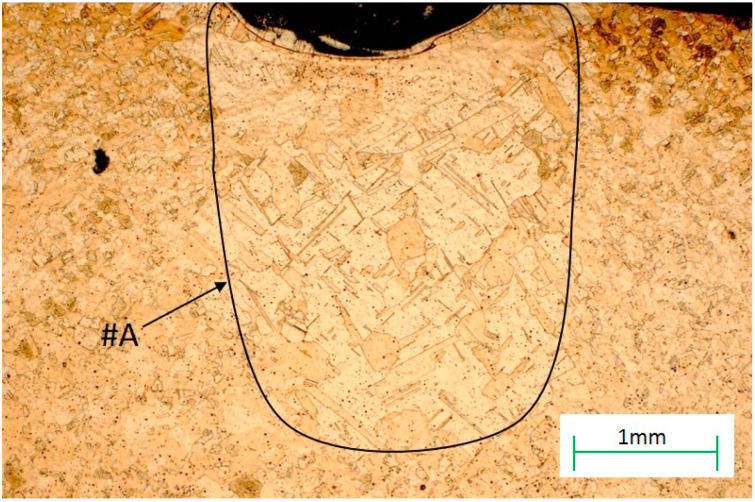
The microstructure of the heat affected zone (HAZ) zone with the recrystallization area, #A, marked.

**Figure 12 materials-13-02430-f012:**
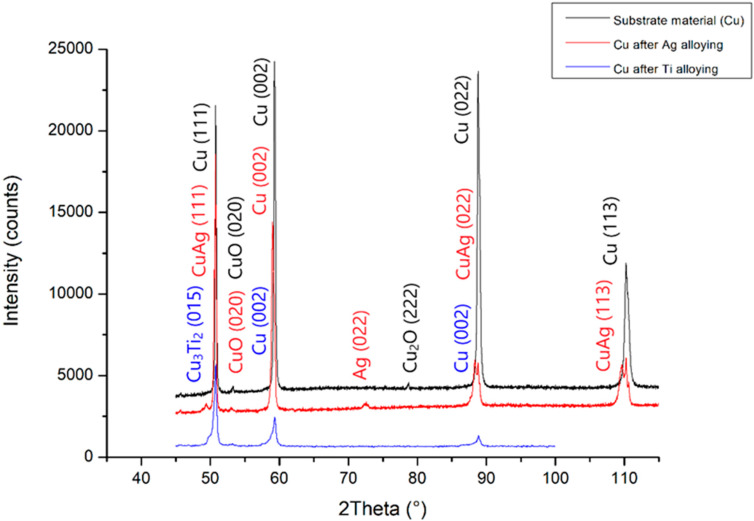
X-ray analysis of the substrate material before and after Ti and Ag alloying.

**Figure 13 materials-13-02430-f013:**
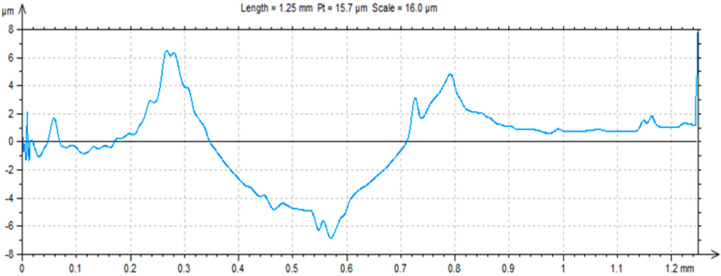
Trace after ball-on-plate test for pure copper.

**Figure 14 materials-13-02430-f014:**
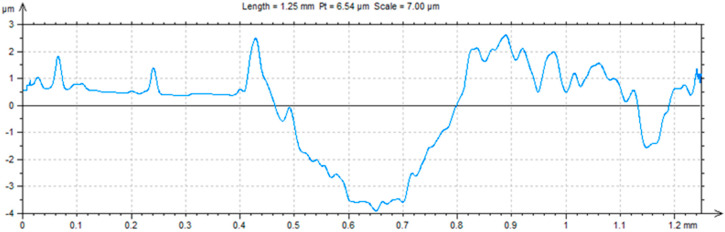
Trace after ball-on-plate test for material (Cu) after Ag alloying.

**Figure 15 materials-13-02430-f015:**
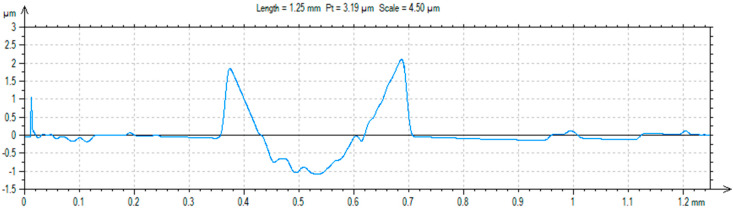
Trace after ball-on-plate test for material (Cu) after Ti alloying.

**Table 1 materials-13-02430-t001:** Process parameters.

Power, kW	Laser Beam Diameter, mm	Scan Number	Scan Speed, m/min	Gas, L/min	Powder
3–4	3	1–3	0.06	Helium, 2	Silver
					Titanium

**Table 2 materials-13-02430-t002:** Results of the EDS spectrum analysis for the areas from Figure 6 (wt.%).

Element	Point #1 (Figure 6a)	Point #2 (Figure 6a)	Point #3 (Figure 6a)
Cu	92.11	31.61	87.32
Ag	7.89	68.39	5.55
O	-	-	7.13

**Table 3 materials-13-02430-t003:** Interplanar distances of the laser-alloyed copper.

The Sample	d, Å
Substrate material	2.08746
Copper after Ag alloying	2.08712
Copper after Ti alloying	2.08690

**Table 4 materials-13-02430-t004:** Averaged results obtained from the hardness measurements using the Vickers method.

The Sample	Average Hardness, HV1	Standard Deviation
Pure copper	112.09	7.65
Copper after Ag alloying	129.30	4.04
Copper after Ti alloying	483.37	2.32

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
