# Peer review of "Microstructure and Properties of the Copper Alloyed with Ag and Ti Powders Using Fiber Laser"

_materials, 2020, doi:10.3390/ma13112430_

Round 1

Reviewer 1 Report

Board comments,

This manuscript is a good study about the increasing of the mechanical properties of the copper without impoverishing its electrical conductivity via laser surface modification. The manufacturing conditions of the coatings as, laser parameters and powders characteristics, were adequate to generate the surfaces with wished features. Microstructures and chemical composition analyses in combination of the electrical and mechanical testing allowed to provide a study of the influence of the microstructural and chemical features on the physicochemical properties of the samples.     

Although the manuscript reaches its aims, it is recommendable to consider the next specific comments.

To add reference in 84 line "component) into solution [REF]."

To replace melting with re-melting in 183 line "segregation of three zones, namely, the melting zone (RZ), the diffusion zone (DZ), and the heat"

Author Response

Thank you very much for the review.

To add reference in 84 line "component) into solution [REF]."

This paragraph was unfortunate, and was to emphasize that ceramic particles are often used for surface strengthening of metals by laser technologies, but very rarely pure metals are used, which cause strengthening due to the formation of intermetallic phases as well as the separation and solid solution mechanism.  Paragraph was moved to the end of the Introduction. Therefore, there is no reference to this paragraph.

To replace melting with re-melting in 183 line "segregation of three zones, namely, the melting zone (RZ), the diffusion zone (DZ), and the heat"

Been changed.

Yours sincerely

Mariusz Krupiński

Reviewer 2 Report

The paper presents an investigation into the microstructure and properties of the copper surface layer treated by laser. The copper was alloyed with Ti and Ag powders. The paper contains some promising results, however, needs severe improvement.

General opinion:

Overall, even the paper is developed, it still has many drawbacks. The gaps in state of the art and the aim of the work are not sufficiently explained and should be improved. The materials and method section is written in a specific form and does not contain sufficient information (the object of the research is not clear enough). Paper structure, namely, the plot of results discussion, should be improved. Especially wear part should be toughly reanalyzed. The whole paper contains no statistics. The same in the conclusions section it should correspond to the aim of the work. And finally, the English language relating to the materials-science should be improved.

Specific comments: 

  • Title: probably authors means "microstructure", please improve. Also, I think that it should be written "powders".
  • Abstract: 'structural studies' rephrase to "microstructural studies"
  • In the abstract, the aim of the work is slightly signalized, while after reading the introduction, it is difficult to understand the idea of the research. - it should be improved.
  • Keywords: the "copper" word is missing, improve the word "structure" to "microstructure". In your future works, please, try to omit the word "structure" while writing about "microstructure" - please take a look at https://www.collinsdictionary.com/dictionary/english/structure
  • L29 - Pure copper can not be strengthening by precipitation etc. (probably authors mean "copper alloy" it must be improved.
  • Please rewrite the L29-L32 - Take look that you mixed "copper" and "copper alloy", it must be specified.
  • The paragraph starting from L44 repeats information about copper recrystallization - should be improved.
  • Grain fragmentation" or "grain refinement" - study if there is any difference and use one of them systematically in the whole text.
  • L99 - boron or boride - please reconsider.
  • In current form introduction does not inform about the drawbacks in knowledge, and does not present clear "aim of the work" - it should be improved.
  • Materials and method section is written "in points" - it is an unusual form of presentation. Also, this section does not explain what the object of the work is, how the samples were prepared (treated - alloyed) and many more likewise:
  • Add sandpaper number # or roughness of the surface. 
  • table 1 - improve the word "titan"
  • how many samples were treated?
  • What was the treated area? Please provide the photo of treated samples (and dimensions of the lasered path)
  • L130: what kind of "tested material" the authors mean? Cross-section, surface, treated - untreated - it should be clearly explained in the paper.
  • XRD paragraph - should be rewritten entirely e.g., "diffraction image"?, what about phase composition? 
  • hardness: HV0.3 with load approx. 5N? The authors must improve that. How many indentations were made, indentations location (a path?) Add statistics. 
  • Was the tribotest reciprocal sliding or wear track was a ring-like? Explain it. 
  • In the whole paper, use "wear track profile" instead of "wipe profile". 
  • How many wear track profile measurements were done? add statistics to compare the wear results. 
  • write "ambient temperature" instead of "room temp."
  • L151-153 - it is not clear what authors mean and why at the beginning of the discussion section? Explain it.
  • Please improve the first area to "re-melting" L155. 
  • Did the surface of the specimens were polished/grinded after alloying? Explain it.
  • L170-171 - the phrase is not precise. What authors mean? Mass or volume? Explain.
  • L174 - must be supported by the literature ref. Alloying can reduce the mech properties, it depends on the chem. element. 
  • rather than "strength properties" use "mechanical properties."
  • in fig 5 - add quantitative results of each spot.
  • l178-181 - EDS supports SEM (or EDS)? please improve the style.
  • L178 - separated or precipitated? Please reconsider the meaning. 
  • Fig7 - where is the surface or DZ? Mark it in the photo.
  • Fig 8 - the same comment as for fig 5.
  • 187 - samples were plastically deformed? It was not mentioned in the "mat & meth" section. The phrase should be improved, especially the last phrase, L190.
  • 197 - graph or plot?
  • What have the results given in table 2 mean? Discuss it.
  • Statistics must be provided in table 3.
  • L210 - improve the word "units". (remove it and add an appropriate unit of hardness). 
  • wipe volume - change to volume wear or volume loss
  • add statistics to the wear results. 
  • What was the tip radius of the profiler? What was the surface roughness before the wear test, was it ground - polished after alloying? Explain. 
  • Conductivity test conditions should be explained in the materials and method section.
  • After improving the manuscript's body, it is advisable to reconsider the conclusions.  

Author Response

Thank you very much for the review.

Reviewer 3 Report

  1. The English level is very frustrating.
  2. Arrows and text should be more contrast and larger (higher) in Figures 5–10, and 12–14.
  3. The “1 mm” dimensional marker will be better in Figure 10.

Author Response

Thank you very much for the review.

Arrows and text should be more contrast and larger (higher) in Figures 5–10, and 12–14.

Figures have been increased as much as possible and arrows in drawings have been corrected.

The “1 mm” dimensional marker will be better in Figure 10.

Corrected. Changes were made as suggested by other reviewers. Figure 10 is now number 11.

Yours sincerely

Mariusz Krupiński

Reviewer 4 Report

The paper is interesting and presents important findings, however it is not publishable in the present form. The paper needs a very deep English editing, for example:

In the text:

"Strengthening of copper materials is most often obtained by grain fragmentation, however this procedure has a disadvantage in the fact that at elevated temperatures, recrystallization occurs, which is not conducive to strengthening".

The statement is obscure and should be re-written.

In the text: "Figure 9. Linear chemical composition analysis at the zone boundary. Measuring line length 85 µm."

It is reasonable to suggest that the authors meant: "Figure 9. Linear chemical composition analysis at the zone boundary. The scale bar is 85 µm". 

In the text: "For samples used after plastic deformation, recrystallization begins with the DZ (diffusion zone) "

It should be "For the samples used after plastic deformation, recrystallization begins with the DZ (diffusion zone)".
 In the text: "Following such treatment, a re-melting zone (RZ), a diffusion (DZ), and a heat affected (HAZ)..."

It should be: "Following such a treatment, a re-melting zone (RZ), a diffusion (DZ), and a heat affected (HAZ) ..."

Very deep scientific editing is necessary.

Author Response

Thank you very much for the review.

The changes that have been made include the comments of other reviewers.

Some descriptions have been changed.

"Strengthening of copper materials is most often obtained by grain fragmentation, however this procedure has a disadvantage in the fact that at elevated temperatures, recrystallization occurs, which is not conducive to strengthening".

Changed to:

Strengthening of copper is most often obtained by plastic deformation, however this method has a disadvantage in the fact that at elevated temperatures, recrystallization occurs, which decreases strengthening.

In the text: "Figure 9. Linear chemical composition analysis at the zone boundary. Measuring line length 85 µm."

It is reasonable to suggest that the authors meant: "Figure 9. Linear chemical composition analysis at the zone boundary. The scale bar is 85 µm".

In the text: "For samples used after plastic deformation, recrystallization begins with the DZ (diffusion zone) "

It should be "For the samples used after plastic deformation, recrystallization begins with the DZ (diffusion zone)".

 In the text: "Following such treatment, a re-melting zone (RZ), a diffusion (DZ), and a heat affected (HAZ)..."

It should be: "Following such a treatment, a re-melting zone (RZ), a diffusion (DZ), and a heat affected (HAZ) ..."

The above comments have been taken into account.

Yours sincerely

Mariusz Krupiński

Round 2

Reviewer 2 Report

Dear Authors,

The paper was toughly improved since I first read it. Thank you for your hard work on improving the quality of the paper submitted to Materials. Now the paper is more suitable for publication. However, I cannot fully accept your response. Particularly, explain which of my questions you cannot answer due to COVID. Mark which questions you mean. Please explain.

Moreover, I think that the COVID is not a reason for publishing the paper with drawbacks or poor quality. 

My comments on the paper.

1. Rev# How many wear track profile measurements were done? The authors explanation is not precise. They wrote: "Abrasion resistance results are average values". My question is "average of what? 1, 2 measurements or more? 

2. Authors: The standard deviation is at 0.000053 for Cu,
427 0.000044 for Cu-Ag, 0.000017 for Cu-Ti. 
Rev# comment: It is not clear and it how it was calculated, provide the raw data. I think that you have made some mistakes during your calculations (too low values). Therefore it still cannot be claimed if there is any differences in quantitative wear results. Especially between pure copper and Ag alloyed results. Please explain.

Moreover, I can interpret your results in that way: contrary to the Ti alloying, the Ag does not improve the wear. Especially the "abrasive surface" area is similar to those of pure copper. And I am not convinced that there is a statistically significant difference between 0.0041 mm3 and 
0.0036 mm3. Please provide valuable evidence of your statement or agree with me and improve the conclusions (regarding wear resistance) and abstract (L22-23).

3. I am worried that authors do not understand the basics of the hardness measurements. Please check in the appropriate ISO standard the relation between the load and the "HV0.3". 

4. Table 4: improve the number "483,374" to 483.37

5. L119: change 'structure' or 'microstructure'

6. this phrase is not clear "The presence of many hard intermetallic phases increases the abrasion resistance of the surface layer laser-alloyed with titanium." should be explained. 

7. This phrase should be improved:  Additionally, the material's abrasion resistance is increased, as a result of the eutectics created following Ag powder alloying and the intermetallic phases created after laser alloying with Ti powder.  - it should be reconsidered if "eutectics" behaves as you claim. Please add the SEM investigations of the wear tack to support your statement.  

8. Also, I maintain my statement that "In current form introduction does not inform about the drawbacks in knowledge, and does not
present clearly "aim of the work" - it should be improved." Where is the novelty of your research - please explain.

Author Response

Dear Reviewer,

Thank you very much for the thorough review

This manuscript is a resubmission of an earlier submission. The following is a list of the peer review reports and author responses from that submission.

Round 1

Reviewer 1 Report

The paper presents the laser treatment of the copper substrate with Ti and Ag chemical elements. The article is focused on the reporting the microstructural phenomena, hardness and wear properties development due to laser processing. 

The object of the study seems interesting though paper contains many drawbacks and should be improved.

First of all, the "extensive editing of the English language and style required". Also, the terminology must be improved, e.g. "chromatic composition", "grinding the grain" and "wipe profile" look unprofessional and I strongly recommend the authors to study the accurate vocabulary/phrases relating to the microstructure and abrasion description. The language, vocabulary and style must be improved. 

Abstract section is blurred. It is challenging to go-though it (improve the style). Authors should prepare it according to Materials requirements.  

The introduction section should be improved and describe the relevance of the object of the study. What is the novelty, please emphasize. At the end of the introduction, add the scientific aim of the study. Now, the objective of the study is not supported by the introduction. The title is too general, and please rephrase it accordingly to the aim of the study. 

Moreover, L56-75 in 75% matches to the reference: Fundamentals of Inductively Coupled Wireless Power .... https://www.intechopen.com/books/wireless-power-transfer-fundamentals-and-technologies/fundamentals-of-inductively-coupled-wireless-power-transfer-systems " and in my opinion this phrase as well as fig. 1 should be removed. Skin effect is known for engineers as well as for scientists and does not need any explanation, especially in the journal as Materials.

Please improve the material and method section. This section is unusually structured; therefore, it is easily found out the drawbacks in methodology. Eg. Provide the laser beam size, dimensions of the treated substrate and area of laser surfaced material or in the end, add necessary information about the wear testing likewise a counterball material (and dimensions), test environment (dry, wet?), "stop condit." probably equals sliding distance? Type of test it is unclear (ball-on-disc/plate?). How many profiles were measured, how the wear was estimated, what was the tip radius. Finally, the wear mechanism should be investigated. 

The graphic material, given in the Results section, should be re-formatted or re-arranged, please improve the figs. 6 and 7 and XRD diffractograms should be presented in one plot (one above another). 

After that, the discussion section should be rewritten to emphasize the scientific idea of the paper and then the appropriate conclusions could be drawn. 

Summing up, the paper should be completely improved and then, checked by a professional English translator.

Reviewer 2 Report

Board comments,

This manuscript could present an appropriate study about the increasing of the hardness and wear resistance of pure copper without impoverishing electrical conductivity of the metallic alloy via laser surface modification with silver and titanium powder. The analyses of the microstructure of laser surface modified samples were carried out through powerful methods, as scanning electron microscopy (SEM), energy dispersive spectroscopy (EDS) and X-ray diffraction spectroscopy (XDS). In addition, the hardness of the samples were adequate evaluated through the correct hardness measurements. Moreover, wear resistance could be correctly analysed via tribological testing and the evaluation of the wear grooves profile through profilometry. Besides, the influence of the powder elements (silver and titanium) about electrical conductivity of samples was studied. The best powder for laser surface modification of the cooper was identified.

This document could reach its aim after the consideration of next comments:

A small introduction about the laser surface modification on copper (e.g. increasing of hardness and reduction of the electrical conductivity) should be added in the abstract part. This could be interesting to reader.  

The words about laser (e.g. laser, laser alloy, laser surface modification and others) should be included in the keywords. This could improve the dissemination of the manuscript.

Suppliers of  the copper bar, laser device, titanium and silver powders should be indicated in the part of the  2.Materials and Methods.

Laser parameters (e.g. laser beam diameter, continuous wave or pulsed, pulse lengths (only pulsed), pulse frequency (only pulsed), beam quality, focused or defocused, Rayleigh length and scan overlapping or scan number) should be detailed in the part of 2.Materials and Methods. This can be of interest to readers.

The details of the copper surfaces (e.g. average roughness and as-received or polished) should be described in 2.Materials and Methods part because this can be interesting to readers.

SEM´s conditions (e.g. potential, current and pressure) should be detailed in the part of the 2.Materials and Methods.

The type of tribological testing (e.g. sliding reciprocating, pin-on-disk, block-on-ball and etc...) should be indicated in the 2.Materials and Methods part.

SEM pictures and EDS of the as-received sample microstructure should be add in the part of the 3.Results and discussion. This could be helpful to evaluate the microstructual change generated by laser surface modification.

Profile images of wear grooves should be included in the part of 3.Results and discussion. These pictures could improve the discussion of the wear resistance field. 

Certain conclusions poor experimental support (e.g.209-211 line "The area of recrystallization depends on the degree of crease, laser power, beam scanning speed, spot width but also the amount and type of powder.") should be removed or, experimental data, which support these conclusions, should be added in the part of 3.Results and discussion.

Specific comments,

To define ATD acronym 41 line "ATD and selection of appropriate heat treatment parameters, modification of the examined alloys"

To add reference 53 line "entirely on the surface of the cable (the so-called Skin effect) [REF]. The epidermal effect for high frequency"

To include reference 117 line "eutectics [REF], what can be seen on the Fig. 4. The EDS analysis in the micro-areas marked on the Fig. 4a" 

To add reference 156 line "the form of eutectics [REF]. The shift of the Cu peak is caused by the dissolution of Ag in Cu. Difference"

To introduce reference 197 line "with titanium [REF]."

Reviewer 3 Report

No comments.